# Microwave Near-Field Dynamical Tomography of Thorax at Pulmonary and Cardiovascular Activity

**DOI:** 10.3390/diagnostics13061051

**Published:** 2023-03-09

**Authors:** Konstantin P. Gaikovich, Yelena S. Maksimovitch, Vitaly A. Badeev, Leo A. Bockeria, Tamara G. Djitava, Tea T. Kakuchaya, Arzhana M. Kuular

**Affiliations:** 1Institute for Physics of Microstructures of Russian Academy of Sciences, Nizhniy Novgorod 603950, Russia; 2Institute of Applied Physics National Academy of Sciences of Belarus, 220072 Minsk, Belarus; olmak0511@gmail.com (Y.S.M.); vitan_bad@mail.ru (V.A.B.); 3Department of Radiophysics and Digital Media Technologies, Belarusian State University, 220030 Minsk, Belarus; 4A.N. Bakulev National Medical Research Center of Cardiovascular Surgery, Moscow 121552, Russia; leoan@bakulev.ru (L.A.B.); tamardjitava@gmail.com (T.G.D.); ttkakuchaya@mail.ru (T.T.K.); lebed8507@mail.ru (A.M.K.)

**Keywords:** medical-biological diagnostics, tomography of lungs, near-field microwave sounding, inverse scattering problems

## Abstract

The developed near-field microwave diagnostics of dynamical lung tomography provide information about variations of air and blood content depth structure in the processes of breathing and heart beating that are unattainable for other available methods. The method of dynamical pulse 1D tomography (profiling) is based on solving the corresponding nonlinear ill-posed inverse problem in the extremely complicated case of the strongly absorbing frequency-dispersive layered medium with the dual regularization method—a new Lagrange approach in the theory of ill-posed problems. This method has been realized experimentally by data of bistatic measurements with two electrically small bow-tie antennas that provide a subwavelength resolution. The proposed methods of 3D lung tomography based on the multisensory pulse, multifrequency, or multi-base measurements are based on solving the corresponding integral equations in the Born approximation. The experimental 3D tomography of lung air content was obtained by the results of the multiple 1D pulse profiling by pulse measurements in several grid points over the planar square region of the thorax. Additionally, the possible applicability of multifrequency measurements of scattered harmonic signals in the monitoring of lungs was demonstrated by four-frequency measurements in the process of breathing. The results demonstrated the feasibility of the proposed control in the diagnosis of some lung diseases.

## 1. Introduction

The proposed electromagnetic tomography is based on the near-field microwave monitoring of lung parameters (depth profiles of air and the blood relative content) in the processes of pulmonary and cardiovascular activity by measured variations in the scattered pulse shape. To date, there are no alternative methods for the dynamical monitoring of these biomedical parameters; thus. the proposed tomography can be applied as a new biomedical diagnostic of certain lung diseases. Algorithms of this dynamical tomography are based on solving the inverse scattering problem, and unlike most of the available methods based on signal imaging, the proposed tomography allows for obtaining quantitative distributions of the physical parameters of the lungs.

By now, studies on microwave diagnostics have mainly been associated with methods of the multisensory tomography of women breast tumors and brain tumors or strokes by measuring signal transmission parameters (e.g., [1,2,3,4,5,6]). A large number of measurement schemes, methods of visualization of corresponding absorbing inhomogeneities, and various measurement systems have been worked out and studied on phantoms and real patients (see in the most competent reviews [7,8]). However, results demonstrate that the quality of the visualization gives little essentially new information compared to the available methods of diagnostics. The conclusion of these reviews stated that “there are very few commercially available and cost-effective microwave-based systems for breast cancer imaging or detection” and “there is still an enormous amount of research and development to be carried out in order to achieve the full capacity of this technology”.

Previous studies on the remote microwave diagnostics of human breathing and heartbeat were initially associated with the task of the radar detection of a human behind the walls of a building or under snow- and landslides, and they were used in some security applications [9,10,11,12,13,14,15,16,17]. In these works, the diagnostics consisted of the visualization (imaging) of signal variations received in the far-field zone that were mainly formed due to refection from the surface of the human body and are primarily related to mechanical vibrations of the skin surface in the process of breathing or heart activity. At that, because of the low penetration at high frequencies, signals scattered from the inner body structure were too small to discern this structure—especially at the poor resolution in the far field. To improve the penetration depth and resolution, as far back as 1979, it was proposed to use near-field contact sensors and lower frequencies (740 MHz–1.5 GHz), and they applied measurements at 915 MHz for the diagnosis of pulmonary edema [18]. Later, with the improvement of sensors, experimental studies at frequencies of 0.5–10 GHz were carried out [19,20], where phase variations of the signal related to breathing and heart activity were observed at the frequency of 0.92 GHz. Recently, measurements using advanced techniques and matched sensors have demonstrated the difference in heart-related signals between healthy people and people after a heart attack, as well as the sensitivity of scattered signals to the pulmonary edema [21].

These results and our experiments stimulated us to develop a new and mathematically consistent diagnostic based on the solution of the inverse scattering problems based on data of bistatic near-field measurements of ultra-wideband (0.05–5 GHz) scattered pulse signals. To realize the high-enough depth resolution of the diagnostics, near-field measurements were used, which, as in near-field optics, provide the necessary subwavelength resolution beyond the Rayleigh limit for deep-penetrating probing signals and sufficiently low frequencies. Such a resolution at low frequencies could not be achieved for far-field measurements, while the use of higher frequencies could only be effective for skin diagnostics because of their strong absorption in living tissues.

The subwavelength near-field diagnostics were successfully applied for the first time in near-field optical microscopy [22]. In our studies of the near-field subsurface microwave diagnostics, it was first used in passive radiothermometry [23,24] (for details see [25]) and later in various methods of the active microwave tomography of subsurface dielectric inhomogeneities [26,27,28,29,30,31,32,33]. In these papers, methods of tomography and holography (for solid targets) of 3D subsurface inhomogeneities of permittivity were worked out and studied experimentally [26,27,28,29,30,31,33]. Additionally, methods of profiling 1D inhomogeneities were developed in papers [30,31]. Unlike far-field radar measurements, where the depth of a scattering element is determined by the signal time or phase delay, the depth sensitivity of near-field methods is provided by other parameters, such as frequencies of harmonic signals [27,28,29,30] (in multifrequency methods); probe aperture sizes [24,25,26] or source-receiver offset [33] (in multi-aperture methods); probe altitudes above the surface [28,33] (in multilevel scanning); and pulse spectrum [31] (in the ultra-wideband pulse sounding).

In this paper, we applied pulse sounding; however, since in our case the pulse length is comparable to the probing depth, we applied not only the pulse spectrum analysis [31] but also a new method based on the analysis of the shape of the scattered pulses itself. Measurements were carried out with the transceiver system that includes two identical bow-tie antennas. The sharp localization of currents on the emitting antenna produces a broad *k*-space spectrum of the probing field that provides a high-depth resolution of subsurface inhomogeneities in the near zone (see Figure A1 in Appendix B). The same antennas were used in our previous studies on the near-field subsurface diagnostics [28,29,30,31,33].

The tomography analysis based on the solution of the nonlinear ill-posed inverse scattering problem is especially complicated in the considered case of time-varying inhomogeneities in the strongly absorbing and frequency dispersive lung tissues under a multilayer thorax cover. In the developed dynamical 1D tomography (profiling) of the air and blood content of the lungs in the processes of breathing and cardiac activity, we developed algorithms based on the dual regularization method, a new powerful method in the Lagrange approach in the theory of nonlinear ill-posed problems [30] that is free from restriction of the perturbation theory. The developed method also includes the retrieval of the multilayer thorax structure that covers lung tissues, where it turned out to be possible to apply another, simpler algorithm. The paper presents the results of a numerical and experimental study of such dynamical tomography. For 3D tomography, where the dual regularization method appeared too resource-consuming, we proposed 3D integral equations obtained in the Born approximation.

Since a multisensory system was not available, we only implemented 1D dynamical tomography (profiling) with the hope that this diagnostic would be further upgraded to 3D tomography. Nevertheless, the first experimental results of 3D lung tomography are also presented here. In this experiment, 3D tomography images were obtained from air content depth profiles in the phase of inhalation retrieved using the 1D algorithm at several points of the 2D thorax surface.

Additionally, the possibility of realizing multifrequency methods based on measuring the complex amplitudes of the scattered harmonic signals was proposed. It should be noted that such measurements could be more informative than measurements of scattered real-valued pulse signals, but it appeared impossible to implement sufficiently fast frequency commutation on the available equipment to obtain data at the required number of frequencies. However, the sensitivity of multifrequency data to variations of the lung structure during breathing is demonstrated here by measurements at four chosen frequencies.

## 2. Materials and Methods

### 2.1. Dynamical Pulse 1D Tomography (Profiling)

#### 2.1.1. Theory

The proposed electromagnetic tomography is based on the near-field microwave monitoring of lung parameters (depth profiles of air- and blood-relative content) in the processes of pulmonary and cardiovascular activity by measured pulse shape variations of the scattered signals. For data, there are no alternative methods for the dynamic monitoring of these biomedical parameters; thus, the proposed tomography can be used as a new biomedical diagnostic of lungs. The algorithms of this dynamical tomography are based on solving the inverse scattering problem and, unlike most existing methods based on signal imaging, allowing for obtaining quantitative distributions of the physical parameters of lungs.

Let us consider a scattering region with the complex permittivity depth profile $\varepsilon (z,T)=\varepsilon_{0i}(z)+\varepsilon_{1}(z,T)$ that consists of the basic multilayer thorax structure of tissues that cover lungs $\varepsilon_{0i}(z)$ and the time-dependent depth profiles of lung permittivity $\varepsilon_{1}(z,T)$ that are determined by variations of air- and blood-relative content profiles $\rho_{air}(z,T)$, $\rho_{blood}(z,T)$ in lungs in the process of breathing and in heart activity. The parameters of the lung are assumed to be unchangeable during the duration of the pulse Δt. The shape of the pulse (the dependence of the received real-valued pulse signal on time *t*) s(t,T)=s[ε(z,T)](t) is determined by profiles ε(z,T). Calculations of the pulse shape by parameters of the lungs are based on the plane wave decomposition of the probing and scattering fields and are described in Appendix B.

The shape of the pulse signal scattered from the probed medium is expressed as the inverse Fourier transform of its frequency spectrum s(ω,T):(1)s(t,T)=∫s(ω,T)exp(iωt)dω,
where ω is the cyclic frequency. Because it would be unsuitable to use such resource-consuming numerical methods as FDTD in pulse calculations, we applied here the approach [28,31] based on the plane wave decomposition of fields radiated by currents generated by a pulse input signal on the surface of a transmitted antenna. It is worth noting here that some problems with such a representation may arise when one solves the inverse problems of restoring inhomogeneities with sharp boundaries (with a derived singularity) because of their smoothing or distortion by the Gibbs effect.

The measured scattered pulse signal can be obtained from complex amplitudes of the scattered harmonic field components $s_{\omega }(\omega ,T)$ generated by the corresponding spectral components of the currents. Using the formulas of this theory given in Appendix B, the received signal can be calculated from the spectrum $s_{\omega }(\omega ,T)$ for any complex permittivity profile as:(2)s(t,T)=Re∫sω(ω,T)exp(iωt)dω,
where Re(*) marks the real part of the integral expression. For brevity, above and below, we used the same symbols for the respective Fourier transforms; thus, they are defined by their arguments. The spectrum of the signal in (1) is expressed as
(3)$$s(\omega ,T)=\frac{1}{2}[s_{\omega }(\omega ,T)+s_{\omega }{\left(-\omega ,T\right)}^{*}].$$

The statement of the inverse problem of the reconstruction of the depth profile ε(z) can be formulated conditions for the equality of the calculated and measured pulse parameters (pulse spectrum or shape). These functional equations are strongly nonlinear and underdetermined; it is impossible to obtain both parts of the complex-valued function of the two variables ε(z,ω) to form parameters of the real-valued scattered pulse. However, it is possible to simplify the statement of the problem by using its specific features.

You can first solve the problem of determining the cover layered structure of the thorax $\varepsilon_{0i}$ and then the problem of restoring the profile of the lung permittivity. To determine $\varepsilon_{0i}$, it is necessary to use measurements in the phase of complete exhalation and diastole ($\rho_{air}(z)=0$, $\rho_{blood}(z)=0$), when the signal variations associated with the heartbeat stop, and the deflated lungs can be considered a homogeneous half-space with the known dielectric parameters. This strongly nonlinear ill-posed problem can be solved using, as a first guess, the available a priory information about this structure, including the known permittivity of tissues of layers [34] and their approximate (or measured by other means) thickness. The depth positions of the interfaces for five layers (skin, fat, muscles, ribs, and lungs) were chosen as unknown parameters. Because the number of these parameters is rather small, this discrete problem is not so ill-posed—it can be solved by minimizing of the quadratic discrepancy between the calculated and measured values of the signal spectrum using the enumerative technique in the reasonable range of layer thicknesses around the first guess *z_i_*_0_:(4)$$\begin{array}{l} \int_{\Delta \omega_{0}}{\left|s[\varepsilon (z_{1i},z_{2j},z_{3k},z_{4l}),\rho_{air}(z)=0,\rho_{blood}(z)=0](\omega )-s_{0}(\omega )\right|}^{2}d\omega \to \underset{D}{\min},\;\\ D\equiv \{z_{1i}=\Delta zi,i=1,..,i_{\max};z_{2j}=\Delta zj,j=i_{\max},..,j_{\max};z_{3k}=\Delta zk,\;k=j_{\max},..,k_{\max};z_{4l}=\Delta zl,l=k_{\max},..,l_{\max}\}, \end{array}$$
where Δz is the step of discretization. To improve the retrieval accuracy, the optimal band of the analysis $\Delta \omega_{o}$ is chosen outside the more high-frequency spectrum of random errors based on the results of numerical simulation. The frequency spectra of the permittivity of the main (unchangeable) layered structure of the thorax are known for all their tissues, including the permittivity of the cover tissues and lungs in phases of complete exhalation and inhalation for all frequencies of the spectrum of the received signal [34].

Then, using the obtained cover lung structure $\varepsilon_{0i}$, it is possible to solve the inverse problems of restoring the profiles of air and blood content in the lungs in the process of breathing and cardiac activity. At the same time, to determine complex-valued variations of the permittivity of the lung profile ε(z,ω,T) from variations of the real-valued parameters of the air $\rho_{air}(z)$ and blood $\rho_{blood}(z)$ content, we used Rayleigh formulas [35] for dielectric mixtures of lung tissues with air and blood:(5)$$\frac{\varepsilon -\varepsilon_{lung}}{\varepsilon +2\varepsilon_{lung}}=\rho_{air,blood}\frac{\varepsilon_{air,blood}-\varepsilon_{lung}}{\varepsilon_{air,blood}+2\varepsilon_{lung}}$$
which leads to the nonlinear expression for the variation of permittivity $\varepsilon_{1}(\rho_{air,blood})=\varepsilon -\varepsilon_{lung}$ in lungs:(6)$$\varepsilon_{1}(\rho_{air,blood})=\frac{\varepsilon_{lung}+2\varepsilon_{lung}\rho_{air,blood}\frac{\varepsilon_{air,blood}-\varepsilon_{lung}}{\varepsilon_{air,blood}+2\varepsilon_{lung}}}{1-\rho_{air,blood}\frac{\varepsilon_{air,blood}-\varepsilon_{lung}}{\varepsilon_{air,blood}+2\varepsilon_{lung}}}-\varepsilon_{lung}.$$

Variations in the scattered signal associated with breathing and cardiac activity are rather small and cannot be seen everywhere on the surface of the chest—especially variations associated with the heartbeat, which are most often recorded in the full inhalation phase, since the blood wave fades in lung capillaries in the exhalation phase. Periodic processes associated with breathing and cardiac activity have different periods; moreover, breathing can be controlled. Therefore, it is possible to isolate the contributions of air and blood to the variations in the permittivity $\varepsilon_{1}$ and solve the corresponding inverse problems separately using the cover structure $\varepsilon_{0i}$, previously obtained from (4).

In this case, the corresponding profiles are represented as $\varepsilon_{1}(\omega ,z)$ = $\varepsilon_{1}[\rho_{air}(z)](\omega ,z)$ or $\varepsilon_{1}(\omega ,z)=\varepsilon_{1}[\rho_{blood}(z),\rho_{air}(z)=\rho_{\mathrm{insp}}](\omega ,z)$, where $\rho_{\mathrm{insp}}$ is the relative air content corresponding to the known value of ε for fully inflated lungs [34].

Let us first consider the statement of these inverse problems based on the spectral approach. The equations of the corresponding inverse problem can be written (omitting the time variable *T*) as the equality condition:(7)$$s[\rho (z)](\omega )=s_{0}(\omega ),\;\omega \in \Delta \omega_{0},$$
where the depth profile $\rho (z)=\rho_{air}(z,T)$ or $\rho (z)=\rho_{blood}(z,T)$, and s[ρ(z)](ω) is the spectrum of the scattered signal calculated from (2), (A18) by the permittivity profile ε(z,ω) obtained for the given profile ρ(z) from (4) and (6); $s_{0}(\omega )$ is the spectrum of the measured scattered pulse. To suppress the effect of random errors, the problem is solved in the informative frequency band $\Delta \omega_{o}$ outside the more high-frequency region of the error spectrum, where the signal exceeds the error level. This band is determined from results of numerical simulation.

To solve this nonlinear ill-posed inverse problem, the dual regularization method [30] was adopted. It consists of minimizing the modified Lagrange function at the simultaneous maximization of the corresponding regularized functional of the dual problem. The modified Lagrange function for this method is written as
(8)$$\begin{array}{ll} L[\rho ](\lambda )= & {\left\|\rho \right\|}^{2}+\frac{1}{\Delta \omega_{0}}\int_{\Delta \omega_{0}}^{}\langle \lambda (\omega ),\left(s[\rho ](\omega )-s_{0}(\omega )\right)\rangle d\omega \\ & +\mu \{{\left(\frac{1}{\Delta \omega_{0}}\int_{\Delta \omega_{0}}^{}{\left|s[\rho ](\omega )-s_{0}(\omega )\right|}^{2}d\omega \right)}^{1/2}+\frac{1}{\Delta \omega_{0}}\int_{\Delta \omega_{0}}^{}{\left|s[\rho ](\omega )-s_{0}(\omega )\right|}^{2}d\omega \}, \end{array}$$
where ${\left\|\rho \right\|}_{L_{2}}^{2}=\frac{1}{\Delta z}\int_{\Delta z}\rho {\left(z\right)}^{2}dz$, $\lambda =(\lambda_{1},\lambda_{2})$, Δz is the region of analysis,μ>0, 〈⋅〉 is a scalar product, $\lambda =(\lambda_{1},\lambda_{2})$, s=(Res,Ims) marks two-dimensional vectors, and μ>0. When the free parameter μ is large enough, the minimum of the modified Lagrange function L[ρ](λ) over ρ(z) exists for certain at any λ. The regularized dual problem is the problem of maximizing the concave functional in the Hilbert space $L_{2}^{2}(\omega_{1},\omega_{2})$ that is expressed as
(9)$$V(\lambda )=\underset{\sigma \in D}{\min}L[\rho ](\lambda )-\alpha \|\lambda {\|}^{2}\to \underset{||\lambda ||\leq \mu }{max},$$ where the maximum is found over λ from the set $\Lambda_{\mu }\equiv \{\lambda =(\lambda_{1},\lambda_{2})\in L_{2}^{2}(\omega_{1},\omega_{2}):\;||\lambda ||\leq \mu \}$, $D=\{\rho \in L_{2}(z_{n},0):0\leq \rho (z)\leq \rho_{\max}\}$. The desired solution is obtained as the saddle point of this process of minimization of (8) with respect to ρ at a simultaneous maximization of (9) with respect to the dual variable λ. The numerical algorithm that implements this method is based on the usual gradient minimization of (8) beginning with the zero first guess when maximizing the functional (9) using a quite obvious explicit expression for its gradient [30].

This method has been studied in numerical simulations and successfully applied to experimental data. However, this algorithm turned out to be too time-consuming for multiple operations in the diagnostics of dynamic processes, in particular due to operations with 2D Lagrange coefficients. To avoid this difficulty, we developed and applied an algorithm based on a functional equation that solves the considered inverse problems in terms of the real-valued shape of the scattered pulse. In this case, solution algorithms with 1D Lagrange coefficients are much less time-consuming and are proven to be suitable for repeated use in diagnosing dynamic processes of breathing and cardiac activity. The corresponding inverse problem equation can be written (omitting the argument *T*) as the equality condition:(10)$$\tilde{s}[\rho (z)](t)={\tilde{s}}_{0}(t),$$
where $\tilde{s}[\rho (z)](t)$ and ${\tilde{s}}_{0}(t)$ are shapes of the modified pulses calculated by the corresponding spectra $s[\rho (z)](\omega ),\;s_{0}(\omega )$ that, such as the spectra in (7), are obtained in the informative frequency band $\omega \in \Delta \omega_{0}$.

When solving this nonlinear equation, the dual regularization method includes 1D Lagrange coefficients that make it much less time-consuming. The desired solution ρ(z) is obtained as a saddle point of the process of minimizing the following modified Lagrange functional:(11)$$L[\rho ](\lambda )={\left\|\rho \right\|}^{2}+\int_{\Delta t}^{}\langle \lambda (t),\left(\tilde{s}[\rho ](t)-{\tilde{s}}_{0}(t)\right)\rangle dt+\mu \{{\left(\int_{\Delta t}^{}{\left|\tilde{s}[\rho ](t)-{\tilde{s}}_{0}(\rho )\right|}^{2}dt\right)}^{1/2}+\left(\int_{\Delta t}^{}{\left|\tilde{s}[\rho ](t)-{\tilde{s}}_{0}(z_{s})\right|}^{2}dt\right)\}$$
over the functional parameter ρ(z) while simultaneously maximizing the corresponding regularized concave functional of the following dual problem:(12)$$V(\lambda )=\underset{\sigma \in D}{\min}L[\rho ](\lambda )-\alpha \|\lambda {\|}^{2}\to \underset{||\lambda ||\leq \mu }{\max}$$
with respect to the 1D Lagrange coefficient λ from the set $\Lambda_{\mu }\equiv \{\lambda \in L_{2}:\;||\lambda ||\leq \mu \}$, $D=\{\rho \in L_{2}(z_{n},0):0\leq \rho (z)\leq \rho_{\max}\}$; Δt is the pulse length. The algorithm of this method was studied in numerical simulation and applied in the experimental study of the proposed dynamical tomography.

#### 2.1.2. Numerical Simulation

As it is well-known, in the considered ill-posed inverse problems, it is impossible to establish a universal relationship between the error parameters (such as the signal-to-noise ratio) and the solution accuracy parameters, which strongly depend on the complexity of the restored function. Each algorithm for solving any ill-posed problem, developed for a specific application, must be studied, tested, and optimized in numerical simulations, taking into account the parameters of the medium under study, the available equipment, measurement errors, and typical and extremal functions expected under the given conditions to be restored.

In Figure 1, the pulse shape and its spectrum are shown for the generator used in the experiments.

Figure 2 shows the optical thickness of the probing signal (integral of the absorption coefficient) in a five-layered living tissue (skin, fat, muscle, ribs, and lungs) for the inhalation and exhalation phases in the signal frequency band. It can be seen that the frequency-depth distribution of the attenuation of the probing field in the lungs is sensitive to the air content in the lungs, at least up to penetration depths of about 10 cm, which provides the informativity of the proposed method.

The numerical simulation of the cover structure retrieval included:

(a)Calculation of the received signal spectrum $s[\varepsilon (z_{1},z_{2},z_{3},z_{4}),\rho_{air}=0,\rho_{blood}=0,](\omega )$ for the simulated five-layer structure in the exhalation phase;(b)Calculation of the corresponding scattered pulse s(t);(c)Adding simulated uncorrelated normally distributed random errors with *rms* that corresponded to that in the measured data;(d)Recalculating the pulse spectrum with errors;(e)Allocation of the informative analysis band $\Delta \omega_{o}$ outside the band of the more high-frequency spectrum of random errors;(f)Solving the inverse scattering problem (4) and comparison of the results with the preset structure $z_{1},z_{2},z_{3},z_{4}$.

The numerical simulation of the $\rho_{air}(z)$ profile retrieval using algorithms (10)–(12) included the calculation of the signal spectrum s[ρ(z)](ω) and pulse shape s[ρ(z)](t) with the cover layer structure retrieved from (4), adding random errors, the recalculation of the spectrum with errors in the analysis band $\Delta \omega_{o}$, the calculation of the modified pulse $\tilde{s}[\rho (z)](t)$, the solution of the inverse problem using algorithms (10)–(12), and a comparison of the results with the preset profiles.

Figure 3 shows the scattered pulse signals and their frequency spectra calculated from the known parameters of the complex permittivity for the inflated and deflated lung tissues. In these calculations, lungs were considered a half-space with constant permittivity under the four-layered cover tissues (skin, fat, muscle, ribs).

As seen in Figure 3, the differences between the pulse shapes calculated for inflated and deflated lungs are rather small, while the corresponding differences in the spectra are more pronounced. Our simulation showed the approximate frequencies at which the spectral contribution of the modeled inhomogeneities disappeared. It turns out that the spectrum of random errors is predominantly located at higher frequencies, which makes it possible to limit the spectral bandwidth of the analysis, eliminating the influence of the corresponding components of these errors. As seen in Figure 3, the informative part of their spectra lies at *f* < 1.5 GHz. The possible effect of the experimental random is investigated by calculating the spectrum of the scattered signal with added random errors at an *rms* value of 1% (relative to the pulse amplitude) that corresponds to the estimated error level of the measurement noise. It was obtained that the spectral errors increased at frequencies *f* > 1 GHz; thus, the informative band appeared even narrower. The bandwidth of 0.05–0.9 GHz was found and used to be optimal in the further processing. To suppress the noise effect during further processing by the pulse signal, it was proposed to recalculate it using the inverse Fourier transform by the informative part of its frequency spectrum and to use the resulting modified pulse $\tilde{s}(t)$ in solving the corresponding inverse problems (4), (10)–(12).

In Figure 4, one can see the results of the numerical simulation of air content profiling (10)–(12) in combination with the reconstruction (4) of the five-layer structure of the permittivity of cover tissues at the *rms* of random errors $\delta_{s}$ = 1%.

The numerical simulation of the 1D tomography presented in Figure 4 was carried out for two preset profiles of air content $\rho_{air}(z)$ in the lungs. The monotonous exponential profile corresponds to the most likely simple profiles during breathing; the Gaussian profile with a sharp maximum simulated a layer of air accumulated under the pleura during pneumothorax. Algorithms (10)–(12) used modified pulses (obtained by recalculating the informative part of their spectrum outside the more high-frequency spectrum of random errors) as the input “measured data”.

In this simulation, a preset structure of the cover tissues $\varepsilon_{0i}(z)$ was used with the inflated lungs (with full inhalation). The same structure was used to calculate the signal parameters in Figure 3. The retrieved structure of the cover tissues was retrieved from the solution of the inverse problem using algorithm (4) from the informative part of the spectrum of the simulated modified signal. In Figure 4e,f, the retrieved and preset real parts of the thorax permittivity profiles $\varepsilon (z)=\varepsilon_{0i}(z)+\varepsilon_{1}[\rho_{air}(z)]$ are shown for the frequency *f* = 250 MHz. In these profiles, the cover structure $\varepsilon_{0i}(z)$ obtained from (4) was combined with lung permittivity $\varepsilon_{1}[\rho_{air}(z)]$ for the Gaussian and exponential profiles $\rho_{air}(z)$ (lines 2 in Figure 4c,d) retrieved from the solution of the inverse problems (10)–(12).

From this simulation, it was found that the rms relative value of respiration-related variations in the scattered signal (informative part) was about 5%; thus, the simulated measurements with errors of 1% are quite informative, especially considering that the applied spectral cleaning significantly suppresses noise effect and, consequently, increases the information content of the input data. The errors in retrieved positions of the boundaries $z_{1},z_{2},z_{3},z_{4}$ determined by (4) ranged from 0.5 mm for $z_{1}$ to δz = 1.5 mm for $z_{4}$. Such restoration accuracy provides a sufficiently high quality of restoration both for a monotonic profile and for a profile with a sharp maximum, as shown in Figure 4.

Based on the presented results, it can be concluded that the proposed algorithms can be effective for lung diagnostics.

### 2.2. Methods of 3D Tomography

The development of 3D subsurface tomography for diagnosing the 3D dynamics of air and blood content in the lungs is a much more difficult task. Its statement can be based on the three-dimensional nonlinear integral relation (A19) between the scattered electric field, the probing field, and the scattered inhomogeneity of the complex permittivity, which was obtained from Maxwell’s equations. In general, even its solution as a direct problem of calculating the scattered field is too resource-intensive, which makes it impossible to use (A19) when solving a 3D inverse problem of retrieving the permittivity with the double regularization method described above. In Appendix B, we propose methods for 3D permittivity tomography based on the Born approximation (A20) of Equation (A19), which, of course, leads to serious limitations of possible applications. This approach can be applied to tomography 3D variations in blood content, since the corresponding variations in the dielectric constant of the lungs are quite small; but in the case of 3D diagnostics of the air composition, the Born approximation could be valid only for a sufficiently small inhalation of the lungs.

Unfortunately, even in the Born approximation (A20), we encountered difficulties in the numerical solution of this 3D integral equation. To overcome these difficulties, we proposed and applied the method based on 2D scanning data with a bistatic transceiver system at a fixed source-receiver offset (base) δr or at δr = 0. Under this condition, Equation (A20) can be represented by the convolution equation with respect to transversal coordinates (A22) that, using Fourier transform with respect to these coordinates, is reduced to the one-dimensional Fredholm integral Equation (A23) for the scattered field in the *k*-space. Using this equation, the corresponding integral equations were obtained for the multifrequency tomography (A24), for the pulse tomography (A27), and for the single-frequency multi-base tomography based on measurements of the scattered signal with the variable parameters δx,δy,δz of the source-receiver base δr (A29). These equations must be multifold solved for each pair of *k*-spectrum components *k*_x_, *k*_y_; then, the 3D complex permittivity is obtained in Cartesian coordinates by the inverse 2D Fourier transform.

It is clear that scanning methods in the 3D tomography of the time-dependent media are inapplicable; thus, it can be realized only with a multisensory system of small enough sensors, where data should be properly measured to reproduce data of scanning at the fixed source-receiver base δr.

Under the conditions of the Born approximation, which can be valid for small variations in the permittivity $\varepsilon_{1}<<\varepsilon$, it is possible to extract the frequency-independent part of $\varepsilon_{1}(\rho )$ from the expression (6). In the air content diagnostics, where $\varepsilon_{air}=1<<\varepsilon_{lung}$, one has
(13)$$\varepsilon_{1}=\varepsilon -\varepsilon_{lung}(\omega )\approx -3\varepsilon_{lung}\rho_{air}/(2+\rho_{air})=-3\varepsilon_{lung}\Phi_{\rho }.$$

The condition $\varepsilon_{1}<<\varepsilon$ can be met for small lung inhalation $\rho_{air}<<1$, which gives us the following linear expression:(14)$$\varepsilon_{1}\approx -\frac{3}{2}\rho_{air}\varepsilon_{lung}.$$

Note that, in this case, it is necessary to use the value of $\varepsilon_{lung}$ for the lungs in the full exhalation phase [34].

In the diagnostics of variations related to the blood content, where $\varepsilon_{blood}\approx \varepsilon_{lung}$, one has $\varepsilon +2\varepsilon_{lung}\approx 3\varepsilon_{lung}$, $\varepsilon_{blood}+2\varepsilon_{lung}\approx 3\varepsilon_{lung}$, and we obtained the required linear expression:(15)$$\varepsilon_{1}\approx \rho_{blood}(\varepsilon_{blood}-\varepsilon_{lung}),$$
where it is necessary to use the value of $\varepsilon_{lung}$ for lungs in the full exhalation phase [34].

Then, using the Born approximations (A24)–(A29) of the nonlinear integral Equation (A19), we obtained the tomography equations for the methods mentioned above, omitting the time variable *T*. The *k*-space integral equation of the multifrequency tomography was obtained from (A24), (14), (15) as:(16)$$s_{\omega }(k_{x},k_{y},\omega )=\int_{z^{\prime }}\rho_{air,blood}(k_{x},k_{y},z^{\prime })K_{mf}^{air,blood}(k_{x},k_{y},z^{\prime },\omega )dz^{\prime },$$
where $K_{mf}^{air}=-\frac{3}{2}\varepsilon_{lung}K_{\omega },\;K_{mf}^{blood}=(\varepsilon_{blood}-\varepsilon_{lung})K_{\omega }$, and $K_{\omega }$ is expressed by (A23).

The *k*-space integral equation of the pulse spectrum tomography was obtained from (A27), (13), (14) as:(17)$$s(k_{x},k_{y},\omega )=\int_{z^{\prime }}\rho_{air,blood}(k_{x},k_{y},z^{\prime })K_{pulse}^{air,blood}(k_{x},k_{y},z^{\prime },\omega )dz^{\prime },$$
where $K_{pulse}^{air}=-\frac{3}{2}\varepsilon_{lung}K,\;K_{pulse}^{blood}=(\varepsilon_{blood}-\varepsilon_{lung})K$, and K is expressed by (A28).

For the air content tomography, it may be more efficient to use a more general Equation (13) that gives us the following equations for multifrequency and pulse tomography:(18)$$s_{\omega }(k_{x},k_{y},\omega )=\int_{z^{\prime }}\Phi_{\rho }(k_{x},k_{y},z^{\prime })K_{mf}^{air}(k_{x},k_{y},z^{\prime },\omega )dz^{\prime },\;K_{mf}^{air}=-3\varepsilon_{lung}K_{\omega },$$
(19)$$s(k_{x},k_{y},\omega )=\int_{z^{\prime }}\Phi_{\rho }(k_{x},k_{y},z^{\prime })K_{pulse}^{air}(k_{x},k_{y},z^{\prime },\omega )dz^{\prime },\;K_{pulse}^{air}=-3\varepsilon_{lung}K$$

One can also propose a method of multi-based single-frequency tomography, in which frequency independence is the main advantage. In this case, the integral equation in the k-space is defined simply by Formula (A29):(20)$$s_{\omega }(k_{x},k_{y},\delta x,\delta y,\delta z,\omega =const)=\int_{z^{\prime }}\varepsilon_{1}(k_{x},k_{y},z^{\prime },\omega =const)[\rho ]K_{\omega }(k_{x},k_{y},z^{\prime },\delta x,\delta y,\delta z,\omega =const)dz^{\prime }$$
with the kernel (A30). Functions $s_{\omega }$ and $K_{\omega }$ are the same as in the multifrequency tomography (15) but at ω=const. The solving algorithm can be based on the signal dependence on any of the base parameters δx,δy,δz. The most serious problem of this method is the difficulty of placing the required number of sensors on the surface of the chest.

Equations (16)–(20) are ill-posed Fredholm integral equations of the first kind that must be multiplied for each pair *k*_x_, *k*_y_, for example, by using the algorithm of Tikhonov’s method of generalized discrepancy in the Hilbert space $W_{2}^{1}$ developed in [27]. The desired 3D structure of the air- or blood-relative content in lungs in Cartesian coordinates should be obtained from the *k*-space solution of corresponding equations using the 2D inverse Fourier transform. For methods (16), (17), it was obtained from:(21)$$\rho_{air,blood}(x,y,z)=\iint \rho_{air,blood}(k_{x},k_{y},z)\exp(ik_{x}x+ik_{y}y)dk_{x}dk_{y}.$$

A similar transform of the *k*-space solutions of (18)–(20) gives their representations in Cartesian coordinates: $\varepsilon_{1}(x,y,z,\omega =const)$ for (18) and $\Phi_{\rho }(x,y,z)$ for (19), (20). Then, it is possible to obtain the desired solution for the studied parameters of the lungs. From (5) and (6) for (18), obtain the following:(22)$$\rho_{air,blood}(x,y,z)=\varepsilon_{1}(x,y,z)/\{[\varepsilon_{1}(x,y,z)+\varepsilon_{lung}]\frac{\varepsilon_{air,blood}-\varepsilon_{lung}}{\varepsilon_{air,blood}+2\varepsilon_{lung}}+2\varepsilon_{lung}\frac{\varepsilon_{air,blood}-\varepsilon_{lung}}{\varepsilon_{air,blood}+2\varepsilon_{lung}}\},$$
and from (13) for (19) and (20):(23)$$\rho_{air}(x,y,z)=\frac{2\Phi_{\rho }(x,y,z)}{1-\Phi_{\rho }(x,y,z)}.$$

Note that some corrections to the results of the Born approximation can be obtained by taking into account the secondary scattering [28].

## 3. Experiment

### 3.1. Dynamical Pulse 1D Tomography (Profiling)

In our experiments, we used the equipment that includes processing, a transceiver system with two identical bow-tie antennas with the length of arms 3.8 cm and the width of 5.4 cm that were used in [28,29,30,33], an oscilloscope GZ10E, broadband (0.05–5 GHz) pulse generators GZ1120ME-01,03 with the pulse time 50 ns at 1024 sweep points (a resolution of about 0.05 ns) and a repetition time of 10 μs, and the control computer for recording signals. The pickup time added up to 10 s (500 pulses). The safety of these measurements in the band of cellular communications was confirmed by the conclusion of the Ethics Committee that was based on estimations based on data of measurements on biological tissues, defining that the upper limit of the mobile phone radiation determined in Europe by the SAR parameter (Specific Adsorption Rate) as 2 W/kg (calculated for 10 g of tissues) more than an order of magnitude exceeds the SAR value estimated for our system radiation.

Variations in the profile of the air content associated with breathing and cardiac activity were rather small and could be observed with our equipment only in some places on the chest; variations in the blood content profile were even smaller and were observed only in the phase of the full inhalation. They stopped in the exhalation phase, when the pulse wave stopped in the capillaries of the lungs. It can be noted that the blood wave propagated only in the capillaries of the lungs and not in the capillaries of different tissues. Of course, the sensitivity of measurements may also depend on the individual depths of the cover tissues. Figure 5 and Figure 6 show the measurement scheme and experimental setup.

The results of the experiments are presented in Figure 7.

These results showed that variations of the received signals in the processes of respiration and cardiac activity were clearly visible; moreover, in more detailed images (Figure 7c,d), these variations significantly exceed random errors, which are also clearly visible. The calculated level of these errors was about 1%; the same level was assumed in the numerical simulation shown in Figure 4. It can also be seen that the changes in signal amplitude during inhalation and exhalation in Figure 7a are similar to those observed in the simulated signals in Figure 3a. The effect of heart activity shown in Figure 7b,d is smaller, but as can be seen in Figure 7d, it significantly exceeded the level of measurement errors.

The images in Figure 7e,f represent variations in air and blood content and were obtained from the measured signal variations in Figure 7a,b using algorithms (4), (10)–(12). They visually demonstrate the processes of the inflation and blood filling of the lungs during breathing and cardiovascular activity (they can be seen even more clearly in the animation given in the *animation.pptx* file included in the Appendix A). Once again, we note that this analysis provides a quantitative diagnosis of the considered physical parameters of the lungs. Thus, the obtained results demonstrate the feasibility of the proposed diagnostics.

### 3.2. Tomography Based on Multi-Position Profiling

In the absence of a multi-sensor system and in the approximation of a smoothly inhomogeneous $\rho_{air}(x,y,z)$ in *x* and *y* directions, we tried to implement the 3D tomography of the lung air content $\rho_{air}(x,y,z)$, applying one-dimensional pulse profiling at 9 grid points over a square *x-y* area with sizes 6 × 6 cm on the chest with the step Δx=Δy=3 cm. Measurements were taken at each point, with breath holding in the inhalation phase. To increase the depth resolution in the transceiver system, smaller bow-tie antennas with an arm length of 2.3 cm and a width of 3.0 cm were used (see in Figure 8).

Due to the small number of point measurements, we used a less resource-consuming 1D algorithm (4, 7–9) for solving the inverse problem from the spectrum of the received pulse signal to obtain the 3D distribution of air content $\rho_{air}(x,y,z)$ in the lungs. Images of such pulse tomography are shown in Figure 9.

One can see in Figure 9 that the observed spatial variations in the structure of the air content $\rho_{air}(x,y,z)$ look realistic. It can be noted that these variations in the x- and y-directions (along and across the direction of the ribs) are comparable; we do not see noticeable effects of anisotropy. This confirms the reliability of the results of dynamical tomography presented in Figure 4. It is important to note that this method, if the conditions of its applicability are met, could be more efficient than the methods described above based on solving three-dimensional equations, since the one-dimensional algorithm based on the powerful dual regularization method is free from the limitations associated with the Born approximation and the nonlinearity of the problem.

### 3.3. Multifrequency Observations of Breathing

As mentioned above, multifrequency measurements can be more informative than pulse sounding. With multi-frequency respiration monitoring, it is necessary to ensure a fast commutation between frequencies that lung parameters could be considered unchangeable during the commutation. Such data can be used in the future when solving the corresponding inverse problem for the implementation of the dynamic tomography of air and blood lung contents.

Unfortunately, it turned out to be impossible with the available equipment to provide sufficiently fast frequency commutation to increase the number of frequencies to the level necessary for solving the corresponding inverse problems of tomography. However, to demonstrate the possibility of such diagnostics, we implemented such measurements of the complex amplitudes of scattered harmonic signals during breathing at four frequencies (2, 3, 4, 5 GHz), which were carried out with the same transceiver system. Sufficiently fast four-frequency commutation was implemented using a variable-frequency oscillator. Data were registered for 10 s with a time step of 10^−5^ s^−1^ and a single-frequency commutation time of 0.01 s so that the total commutation time was 0.04 s. Figure 10 shows the results of measurements during breathing.

The results show that the measurements are sensitive to lung transformation during respiration; moreover, this transformation is very uneven at different frequencies. This sensitivity indicates the possibility of using these measurements for subsurface multifrequency lung diagnostics in 1D profiling based on (4) and (8) or in 3D tomography (16) and (18).

## 4. Discussion

Let us compare the developed dynamic tomography with the existing methods of subsurface diagnostics and consider the areas of possible biomedical application of this microwave tomography. Available methods for diagnosing pathologies in various diseases include magnetic resonance imaging (MRI), radiography, fluoroscopy, X-ray computed tomography (CT), and various types of ultrasound computed tomography (USCT) [36]. These methods are based on the use of various waves (electromagnetic, acoustic) capable of penetrating the human body and being transformed by the inhomogeneities of its tissues. The measured parameters of transmitted, reflected, or scattered waves provide valuable information about these inhomogeneities. However, all available methods have fundamental limitations.

X-ray images visualize the transverse 2D distributions of ray absorption in the human body (integral of the absorption coefficient along the ray). In these images, the shadows from the near-surface layers of human tissues are superimposed on the shadows from deeper layers, making them difficult to interpret. Three-dimensional CT images showed the extinction coefficient distribution (depending mainly on tissue density) in each cross section of the body, which visualizes the structure of living tissues. This method is much more informative than the method of simple X-ray diagnostics; however, it can damage DNA, is technically very complex, and is not free from artifacts. Unlike X-ray methods, our microwave diagnostics are not capable of damaging DNA; a probing signal with a spectrum in the cellular frequency band has a much lower thermal effect than mobile phones. Additionally, this ultra-wideband pulsed signal is not capable of causing any resonant effects in the DNA of living tissues. The safety of the measurements was confirmed in the supplied conclusion of the Ethics Committee.

Ultrasonic computed tomography (USCT) methods use high-frequency sound reflections from tissue interfaces and scattering from density inhomogeneities in the human body. The ability to select the depth in the reflection signal allows you to obtain the tomographic visualization of the tissue structure with a resolution determined by the choice of frequency. In addition, Doppler measurements make it possible to observe and measure currents in blood vessels. The information in conventional USCT data is mostly qualitative and is more difficult to express in physical terms than in other tomography methods; thus, its interpretation requires qualified service personnel.

Summing up, we can say that all of the above methods are based mainly on signal visualization and provide qualitative information about the tissues of the human body, which requires experienced highly qualified medical personnel for interpretation, as well as additional data (symptoms, blood tests, etc.). Our diagnostics provided quantitative information about the physical parameters of the lungs, which is unattainable for these methods.

In addition, few of the available methods are suitable for monitoring lung parameters, such as the recently developed real-time MRI (magnetic resonance imaging) [37]. The MRI method, based on the measurement of radio emission during the spin-orbit transition of hydrogen atoms in a magnetic field, visualizes the distribution of hydrogen atoms (mainly in water molecules) in living tissues, especially in aqueous fluids such as blood and lymph. In Figure 11, two screenshots of thorax from the computer playback of the web site video [38] with the animation of sectional images in the real-time MRI (Figure 11a,b) were given in comparison with the screenshots from the animation of our lung tomography (Figure 11c,d) (see also in the *animation.pptx* file in Appendix A).

As is easily seen in the MRI images (Figure 11a,b), the only visible objects in lungs are small white spots that correspond to the cross-sections of the largest blood vessels. They appeared in the systole (Figure 11a) and disappeared in the diastole (Figure 11b), where it is also possible to see the shift of the diaphragm position at breathing. In Figure 11c,d, where screenshots of our dynamical tomography are demonstrated, one can see that our tomography gives lung parameters that are unattainable by the real-time MRI. Here, it should be emphasized once again that, unlike MRI and other methods based on tomography (signal imaging), our tomography gives quantitative values of the physical parameters of the lungs.

Let us consider the pathologies in which the developed tomography can be applied. One of them is pneumothorax (the concentration of air and other gases in the pleural cavity), which can occur spontaneously (primary) or as a consequence of lung disease or chest trauma (secondary). The possibility of the proposed microwave diagnostics of such air layers in the lungs is demonstrated by numerical simulation in Figure 4c,d. Additionally, our method can be used to control operations of artificial pneumothorax, the introduction of air into the pleural cavity to obtain the collapse (atelectasis) of the affected lung.

Atelectasis (partial or complete pulmonary collapse of the lung) may be associated with a variety of lung pathologies. Primary atelectasis is neonatal atelectasis, in which the lungs remain unexpanded; secondary atelectasis may be associated with various lung diseases such as pneumonia, tumors, pulmonary shocks, pleural emphysema, lung injury (pneumothorax, hemothorax), hydrothorax, foreign body aspiration, or other pathologies. Corresponding strong distortions of air exchange profiles and their dynamics can be detected by the developed microwave method.

Some pulmonary pathologies are associated with the concentration of fluid in the lungs during pulmonary edema (blood plasma transudate from capillaries) associated with certain cardiovascular diseases of the lungs or with traumatic injuries [39]. Very similar symptoms can be associated with hydrothorax—the accumulation of fluid (transudate) in the pleural cavity. This may appear with cardiac decompensation. Pleural effusion can also be associated with hemothorax (blood concentration) or after cardiac surgery [39]. The rapid development of such pathologies requires constant monitoring, which can be provided by the proposed microwave diagnostics of the dynamics of air and blood content in the lungs.

Of great interest may be research on microwave diagnosis and the monitoring of COVID-19 pulmonary pathologies and their consequences. During the pandemic, CT imaging proved to be the most effective in diagnosing relevant lung pathologies that appear on CT images as “frosted glass” spots. However, long-term use of this method could be associated with exposure to X-rays. MRI diagnostics turned out to be significantly less effective and are used only as a background study. In this regard, it seems appropriate to study the possibility of using the developed method in the diagnosis of pathologies associated with COVID-19 [40,41].

## 5. Conclusions

In this work, the method of dynamical lung tomography was developed and studied both in numerical simulations and experimentally. The proposed near-field microwave pulse diagnostics made it possible to obtain information about the depth profiles of the relative content of air and blood in the lungs during breathing and the heart beating. This information is unattainable by other available methods based mainly on signal visualization (imaging), while the proposed tomographic analysis gives the physical parameters of the lungs. The physical novelty of this study lies in the recovery of the desired parameters from the shape of the scattered pulse from the solution of a nonlinear ill-posed inverse problem in the extremely complicated case of a highly absorbing and frequency-dispersive layered medium. The solution algorithms are based on the double regularization method in the Lagrangian approach of the theory of ill-posed problems, a new method that is free from the limitations of perturbation theory. They involve the shape of the scattered pulse or its frequency spectrum as input into the analysis. The proposed near-field tomography was experimentally implemented based on bistatic measurements of scattered UWB (0.05–5 GHz) pulsed signals using a transceiver system based on electrically small bow-tie antennas providing subwavelength resolution. The retrieved dynamics of the air and blood content profiles during breathing and cardiac activity were demonstrated.

In the Born approximation, integral equations were obtained that generalized the theory of three-dimensional microwave tomography to the cases of pulse and multifrequency measurements. In the absence of a multi-sensor system, we obtained and demonstrated in this paper 3D lung tomography derived from air content profiles obtained by measurements at multiple points on a square chest area in the inhalation phase.

Additionally, the possibility of implementing such tomography based on the data of multifrequency measurements of the complex amplitudes of scattered harmonic signals during four-frequency measurements of breathing was experimentally studied. The results show that measurements are sensitive to the lung transformation; therefore, such measurements could be applied to the subsurface multifrequency diagnostics of lungs. To perform this, it is necessary to increase the number of frequency channels to the level required to solve the corresponding inverse problems of tomography. Unfortunately, we failed to provide sufficiently fast frequency commutation on the existing equipment.

The possibility of using the proposed tomography for the medical diagnosis of certain lung diseases was discussed. Of course, the results shown in this article are very preliminary and are still very far from real applications in medical diagnostics. Individual variability of the structure and permittivity of tissues should be studied and taken into account (see, for example, [42]). We also hope that the measurement schemes, instrumentation, and sensitivity will be substantially optimized. However, these first results are encouraging and should stimulate further research.

Summarizing, we conclude that the presented results of modeling and experimenting demonstrate the possibility of using the developed dynamical tomography in the biomedical diagnostics of lung parameters that are unattainable in diagnostics by other known methods. These parameters are very informative in the diagnosis of pulmonary edema, pneumo- and hydrothorax, and atelectasis pneumonia and its consequences. The possible applicability of this new tomography can also be considered for the diagnosis of cardiopulmonary pathologies of COVID-19.

## Figures and Tables

**Figure 1 diagnostics-13-01051-f001:**
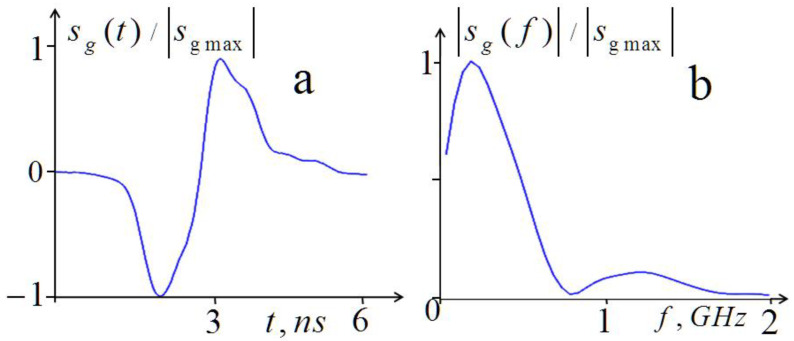
Shape of the generator pulse (**a**) and its frequency spectrum (**b**).

**Figure 2 diagnostics-13-01051-f002:**
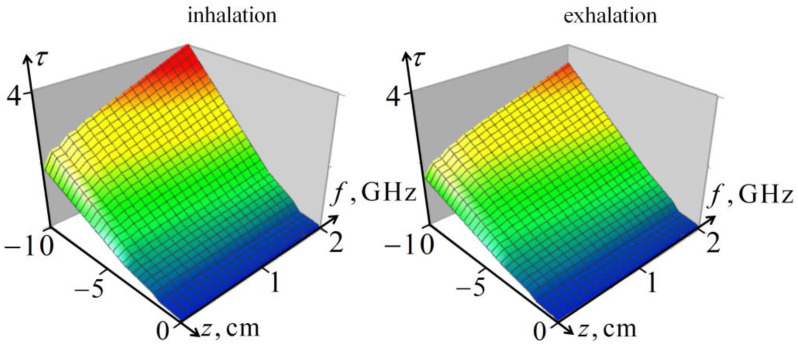
Frequency-depth distribution of the optical thickness τ in five-layered tissues of the thorax.

**Figure 3 diagnostics-13-01051-f003:**
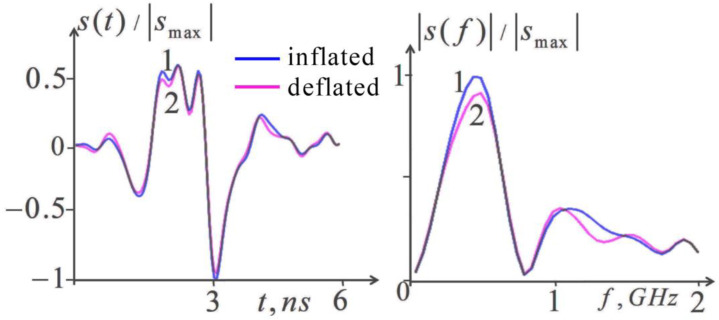
Numerical simulation of the lung inflation effect. Left, calculated received pulses; right, corresponding frequency spectra; 1, inflated lungs; 2, deflated lungs.

**Figure 4 diagnostics-13-01051-f004:**
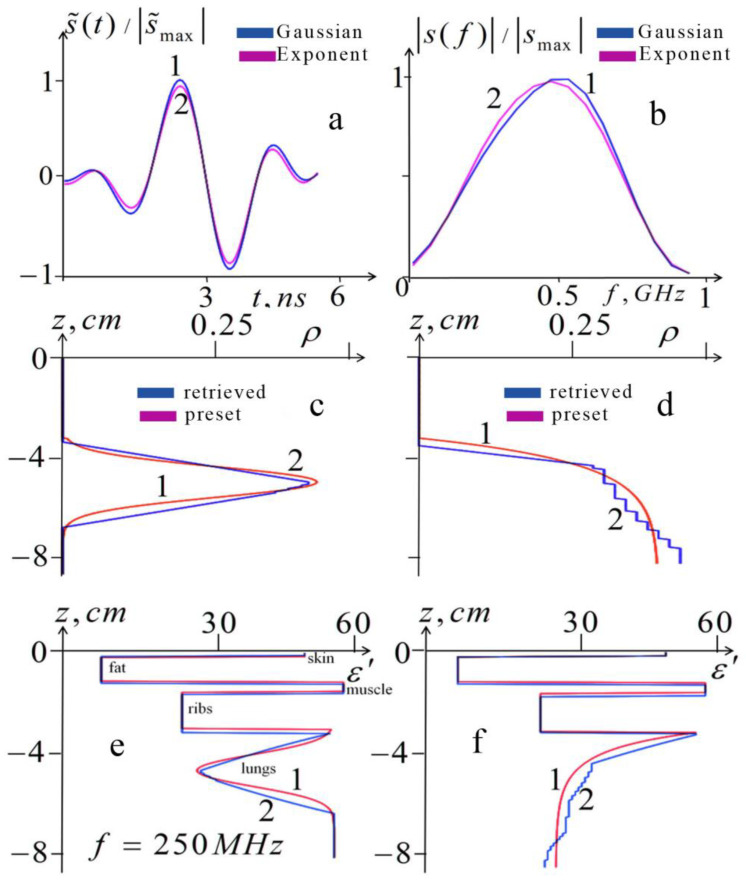
Numerical simulation of 1D tomography (profiling) of the lung air content and permittivity. The top row demonstrates the simulated “input data”: (**a**)—modified pulse signals; (**b**)—frequency spectra of modified signals. Line 1 corresponds to the simulated Gaussian profile (line 1 in Figure 4c); line 2 corresponds to the exponential profile (line 1 in Figure 4d). The middle row shows simulated (lines 1) and retrieved (lines 2) lung air content profiles $\rho_{air}(z)$: (**c**)—Gaussian preset, and retrieved from the data in Figure 4a (line 1) profiles; (**d**)—exponential preset and retrieved from the data in Figure 4a (line 2) profiles. The lower row shows the preset (lines 1) and retrieved (lines 2) profiles of the real part of the thorax permittivity $\varepsilon^{\prime }(z)$ at a frequency *f* = 250 MHz, which combine the structure of the chest cover reconstructed from (4) and the permittivity profiles in the lungs: (**e**)—profiles correspond to the Gaussian profiles in Figure 4c; (**f**)—profiles correspond to the exponential profiles in Figure 4d.

**Figure 5 diagnostics-13-01051-f005:**
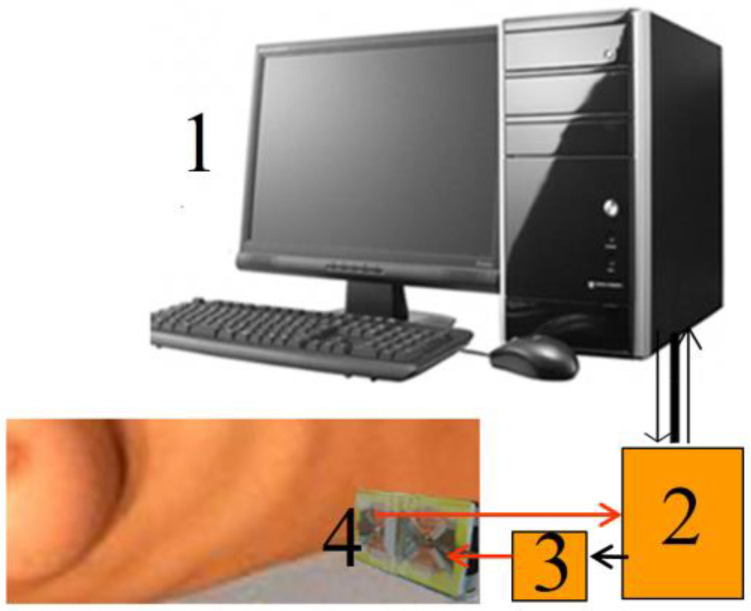
Measurements scheme: 1, computer; 2, oscilloscope; 3, generator; 4, receiving and transmitting antennas.

**Figure 6 diagnostics-13-01051-f006:**
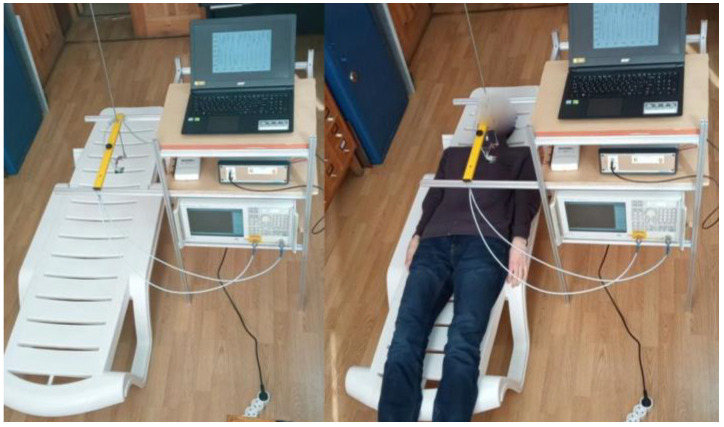
Experimental setup.

**Figure 7 diagnostics-13-01051-f007:**
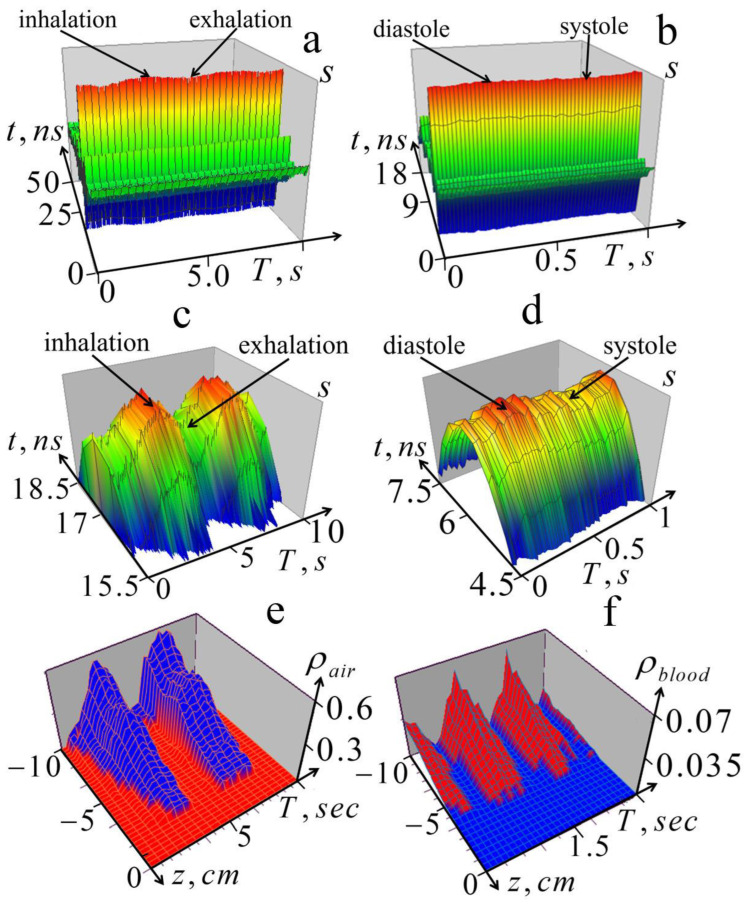
Experimental results of the dynamic 1D tomography (10)–(12) of variations in profiles of the relative air and blood content during breathing and cardiac activity. Experimental results of the dynamic 1D tomography (10)–(12) of variations in profiles of the relative air and blood content during breathing and cardiac activity. Upper row: (**a**)—variations of the received pulse signal *s*(*t*,*T*) associated with breathing; (**b**)—variations of the signal associated with cardiac activity (in the phase of full exhalation). Middle row: (**c**)—the upper part of the pulse shown in Figure 7a; (**d**)—the upper part of the pulse shown in Figure 7b. Bottom row: (**e**)—retrieved dynamics of the air content profile $\rho_{air}(z,T)$ in the process of breathing; (**f**)—retrieved dynamics of the blood content profile $\rho_{blood}(z,T)$ during heartbeat.

**Figure 8 diagnostics-13-01051-f008:**
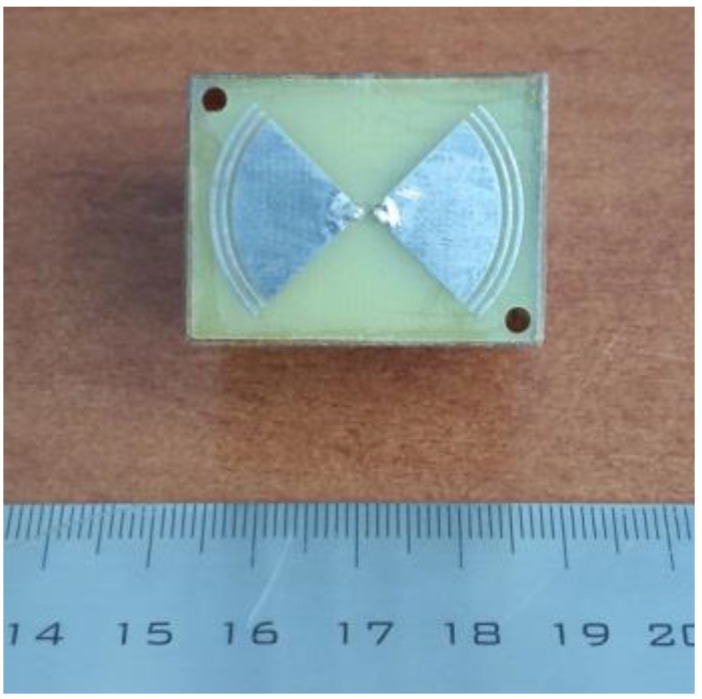
Bow-tie antenna.

**Figure 9 diagnostics-13-01051-f009:**
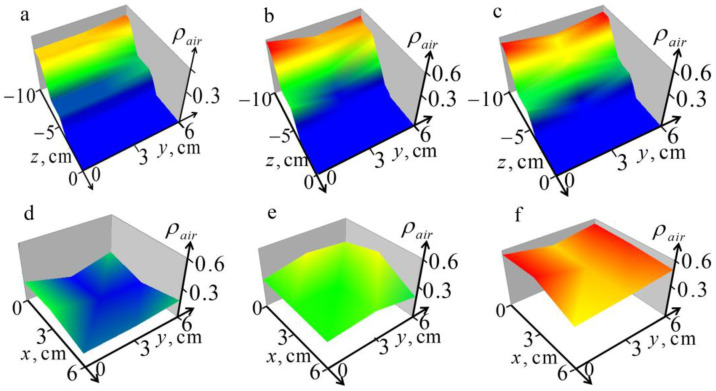
Tomography sections of the retrieved 3D air content distribution in lungs: (**a**–**c**) images in the vertical sections at *x* = 0, 3, 6 cm; (**d**–**f**) images in the horizontal sections *z* = −5, −7, −9 cm.

**Figure 10 diagnostics-13-01051-f010:**
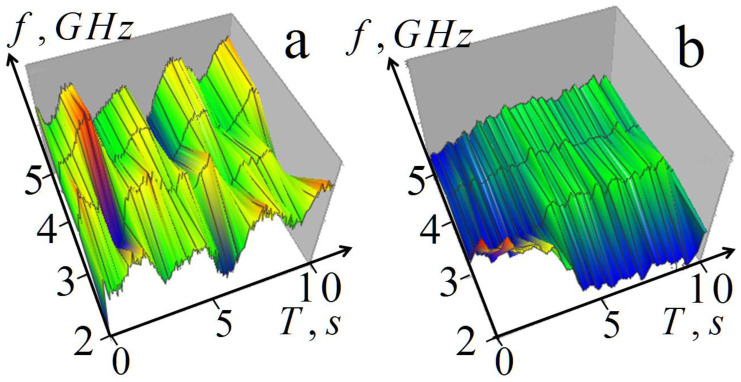
Four-frequency measurements of scattered signal amplitude variations during breathing: (**a**) free breathing; (**b**) exhalation followed by breath-holding.

**Figure 11 diagnostics-13-01051-f011:**
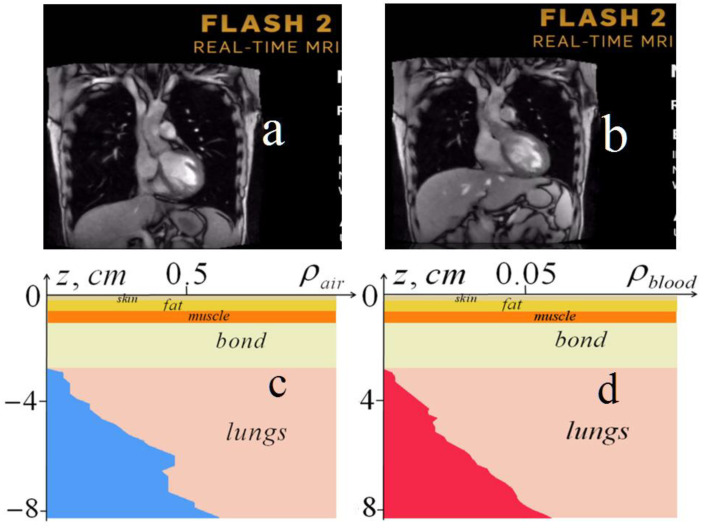
**Upper** row: screenshots of the thorax cross-sections from the MRI animation: (**a**) in phases of inhalation and systole; (**b**) in phases of exhalation and diastole. **Bottom** row: screenshots of the developed microwave dynamical tomography: (**c**) depth profile of the relative air content in the phase of inhalation; (**d**) depth profile of the relative blood content in the phase of systole.

## Data Availability

No new data were created.

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
