# Peer review of "Microwave Near-Field Dynamical Tomography of Thorax at Pulmonary and Cardiovascular Activity"

_diagnostics, 2023, doi:10.3390/diagnostics13061051_

Round 1

Reviewer 1 Report

The authors present a novel approach for microwave tomography of thoracic activity. The work needs an English grammar review (spelling mistakes, wrong use of articles, etc).

Comments are included in the PDF attached.

Main concern:

Not in the paper and not in the IRBS is there a reference to the safety regarding limits on humans' exposure to microwave radiation. Even though there is not one single consensus, the authors should abide by at least one protocol or standard.

Reviewer 2 Report

Review Manuscript ID: diagnostics-2211102
Type of manuscript: Article
Title: Microwave Near-Field Dynamical Tomography of Thorax at Pulmonary and Cardiovascular Activity
Authors: Konstantin P. Gaikovich*, Yelena S. Maksimovitch, Vitaly A. Badeev, Leo A. Bockeria, Tamara G. Djitava, Tea T. Kakuchaya, Arzhana M. Kuular
Journal: Diagnostics
Publisher: MDPI Basel Switzerland
Submitted to the section: Medical Imaging and Theranostics, Advances in Cardiovascular and Pulmonary Imaging
Date: 7 February 2023

The manuscript covers both theoretical and experimental aspects of thorax tomography during both cardiovascular and pulmonary activities. The following provides some remarks.

* Regarding solving integral equations, it is unclear whether those equations were obtained using the Born approximation or they were solved using the Born approximation. The phrases containing these expressions seem ambiguous and different readers could interpret them differently. Please clarify.

* Do you mean frequency computation or frequency commutation? This is also confusing as both expressions can mean very different things.

* The organization of the paper seems to be missing.

* Some figures have low qualities due to low resolutions, at least that is what appears in the printed version. They are a little bit blurred. Improving low-quality figures, such as making them sharper would be better for the article.

* Confirm that Re denotes the real part of the corresponding (integral) expression.

* What guarantees that rho_0 is never zero?

* The integral in (17A) is missing the domain. Are you integrating over a ball in R^3, over the whole space R^3 itself, or something else? This is also unclear to many readers.

* Provide a comma after e.g.

* [11,12] (use a comma).

* see [17] for more details (wrong bracket position).

* could not

* A possible suggestion for an improved abstract.

The developed near-field microwave diagnostics of the dynamical lungs tomography provides information about variations of air and blood content depth structure in processes of breathing and heart beating that is unattainable for other available methods. The method of dynamical pulse 1D tomography (profiling) is based on solving the corresponding nonlinear ill-posed inverse problem in the extremely complicated case of strongly absorbing frequency-dispersive layered medium with the dual regularization method – a new Lagrange approach in the theory of ill-posed problems. This method has been realized experimentally by data of bistatic measurements with two electrically small bow-tie antennas that provide a subwavelength resolution. Proposed methods of 3D lung tomography based on the multisensory pulse, multifrequency or multi-base measurements are based on solving corresponding integral equations in the Born approximation. Experimental 3D tomography of the lung air content has been obtained by results of the multiple 1D pulse profiling by pulse measurements in several grid points over the planar square region of the thorax. Also, the possible applicability of multifrequency measurements of scattered harmonic signals in the monitoring of lungs has been demonstrated by four-frequency measurements in process of breathing. The results demonstrated the feasibility of the proposed control in the diagnosis of some lung diseases.

Round 2

Reviewer 1 Report

The authors have thoroughly revised their manuscript. I appreciate their answers very much.

There are two minor comments that the authors should address.

* The comment of Line 145 (of the original draft) - The limitations of the plane-wave approach that the authors correctly answered in their response letter, should also be included in the final draft.

* The comment of Line 159 (of the original draft) - the gauss function is not limited (thus, not time limited and not frequency limited). However, it is clear that in practical appreciation, one can consider them "almost" limited. That was the main intention of my question - please clearly define what you call delta_omega (the frequency band). Is it -3 dB? is it -40 dB?

After these comments are taken care of, the manuscript will be ready for publication in this journal.

Reviewer 2 Report

Second review Manuscript ID: diagnostics-2211102
Type of manuscript: Article
Title: Microwave Near-Field Dynamical Tomography of Thorax at Pulmonary and Cardiovascular Activity
Authors: Konstantin P. Gaikovich*, Yelena S. Maksimovitch, Vitaly A. Badeev, Leo A. Bockeria, Tamara G. Djitava, Tea T. Kakuchaya, Arzhana M. Kuular
Journal: Diagnostics
Publisher: MDPI Basel Switzerland
Submitted to the section: Medical Imaging and Theranostics, Advances in Cardiovascular and Pulmonary Imaging
Date: 23 February 2023

Thank you for revising your manuscript. The following provides some additional remarks.

* Perhaps equation (6) can be written in fraction form, with the long horizontal line as a divider between the denominator and numerator.

* In Figure 3, perhaps the horizontal line could take a different color instead of blue. Furthermore, distinguishing the two curves with solid and dashed would help readers who have no color printer. Similar remarks with other figures.

* Why do you remove the captions of Figures 5 and 6?

* Line 1069: const should not be italics.

* For high-quality typesetting, consider using LaTeX instead of Microsoft Word.
